



**Rapidly increasing sulfate concentration: a hidden promoter of eutrophication in**
**shallow lakes**
Chuanqiao Zhou[a,1], Yu Peng[a,1], Li Chen[a], Miaotong Yu[a], Muchun Zhou[b], Runze Xu[a],
Lanqing Zhang[a], Siyuan Zhang[c], Xiaoguang Xu [a,*], Limin Zhang[a], Guoxiang Wang[a]
[a] School of Environment, Nanjing Normal University, Jiangsu Center for Collaborative
Innovation in Geographical Information Resource Development and Application,
Jiangsu Key Laboratory of Environmental Change and Ecological Construction,
Nanjing 210023, China
[b] China Aerospace Science and Industry Nanjing Chenguang group, Nanjing 210022,
China
[c] School of Energy and Environment, Southeast University, Nanjing 210096, China
*Corresponding author. 1, Wenyuan Road, Xianlin University District, Nanjing,
210023, China
E-mail address: xxg05504118@163.com
[1] Both authors contributed equally
**Keywords:** Sulfate reduction; iron reduction; phosphorus release; eutrophication;
sulfate reduction bacteria
**Abstract:**
Except for excessive nutrient input and climate warming, the rapidly rising $SO_4^{2-}$
concentration is considered as a crucial contributor to the eutrophication in shallow
lakes, however, the driving process and mechanism are still far from clear. In this study,
we constructed a series of microcosms with initial $SO_4^{2-}$ concentrations of 0, 30, 60, 90,





120 and 150 mg/L to simulate the rapidly $SO_4^{2-}$ increase of Lake Taihu subjected to
cyanobacteria blooms. Results showed that the sulfate reduction rate was stimulated by
the increase of initial $SO_4^{2-}$ concentrations and cyanobacteria-derived organic matter,
with the maximal sulfate reduction rate of 39.68 mg/L·d in the treatment of 150 mg/L
$SO_4^{2-}$ concentration. During the sulfate reduction, the produced maximal $\sum S^{2-}$
concentration in the overlying water and acid volatile sulfate (AVS) in the sediments
were 3.15 mg/L and 11.11 mg/kg, respectively, and both of them were positively
correlated with initial $SO_4^{2-}$ concentrations ($R^2$=0.97; $R^2$=0.92). The increasing
abundance of sulfate reduction bacteria (SRB) was also linearly correlated with initial
$SO_4^{2-}$ concentrations ($R^2$=0.96), ranging from $6.65×10^7$ to $1.97×10^8$ copies/g. However,
the $Fe^{2+}$ concentrations displayed a negative correlation with initial $SO_4^{2-}$
concentrations, and the final $Fe^{2+}$ concentrations were 9.68, 7.07, 6.5, 5.57, 4.42 and
3.46 mg/L, respectively. As a result, the released TP in the overlying water, to promote
the eutrophication, was up to 1.4 mg/L in the treatment of 150 mg/L $SO_4^{2-}$ concentration.
Therefore, it is necessary to consider the effect of rapidly increasing $SO_4^{2-}$
concentrations on the release of endogenous phosphorus and the eutrophication in lakes.
**1.Introduction**
Nowadays, cyanobacteria bloom in eutrophic lakes has become one of the most
serious problems in freshwater lakes all over the world (Iwayama et al., 2017; Ho et al.,
2019). Phosphorus, as a necessary nutrient for biological growth, is considered to be
one of the main limiting factors of lake eutrophication (Ni et al., 2020). In recent years,
the input of exogenous phosphorus has been effectively controlled, while the release of



endogenous phosphorus is still an urgent problem in eutrophic lakes (Liu et al., 2018;
Guo et al., 2020). The release of endogenous phosphorus is affected by many factors,
such as wind and wave and the cyanobacteria decomposition (Xu et al., 2018; Zhao et
al., 2019). There are many forms of phosphorus in freshwater lake sediments, including
aluminum bound phosphorus (Al-P), iron bound phosphorus (Fe-P), etc. Among them,
Fe-P, formed under the condition of high dissolved oxygen (DO), is the most active
form of phosphorus in the sediments, which has a more obvious response to the change
of DO (Zhang et al., 2020). The accumulation and decay of cyanobacteria in eutrophic
lakes will change the physical and chemical environments of water body and form
anaerobic reduction conditions (Yan et al., 2017). This will facilitate the reduction of
iron oxides and lead to the desorption and release of Fe-P in sediments, resulting in the
increase of endogenous phosphorus release (Zhao et al., 2019).

Iron reduction plays an important role in natural ecosystems. It has been reported

that dissimilatory reduction of iron accounts for 22% of the total amount of organic
matter anaerobic mineralization in offshore areas (Thamdrup et al., 2004). According
to the classical theory, iron oxides or hydroxides can adsorb phosphorus in the water
and form Fe-P precipitation (Gunnars et al., 1997). In freshwater lakes, the lack of Fe(III)
content or the diagenesis of organic phosphorus may be the reason for the lack of
phosphorus in the overlying water. Therefore, the formation of iron oxides on the
surface of sediments is closely related to the phosphorus cycle process (Amirbahman
et al., 2003; Chen et al., 2014). The interaction between iron and phosphorus is reflected
in the effect of adsorption and desorption of Fe oxide on the P content in the overlying



water, since Fe-P is the main internal source of phosphorus (Wu et al., 2019). Iron
oxides can be used as both the source and destination of phosphorus in lake ecosystems
(Mort et al., 2010; Azam et al., 2014). In anaerobic reduction environments, iron
reduction can significantly promote the resolution of Fe-P. The $Fe^{2+}$ generated by the
reaction can form FeS solid with soluble sulfide. In addition, free $Fe^{3+}$ will combine
with humus to form stable complex, which further prevents the co-precipitation process
of phosphorus and iron oxides (Mort et al., 2010; Zhang et al., 2020). Therefore, iron
reduction process driven by cyanobacteria decomposition affects the circulation of
phosphorus in freshwater lakes.

Due to the $SO_4^{2-}$ concentration in seawater reaching 28 mM, sulfate reduction

process with the participation of sulfate reduction bacteria (SRB) has received
considerable attention in the basic material cycle of marine biogeochemistry (Fike et
al., 2015; Pan et al., 2020). In freshwater lakes, the $SO_4^{2-}$ concentration is less than 800
μM, which is generally considered insufficient for continuous sulfate reduction (Hansel
et al., 2015). However, in recent years, with the increasing $SO_4^{2-}$ concentration in
freshwater lakes, the impact of sulfate reduction on the material cycle of lake
ecosystems may be far beyond our knowledge (Dierberg et al., 2011; Baldwin et al.,
2012; Yu et al., 2013). In the past 70 years, the $SO_4^{2-}$ concentration in Lake Taihu has
increased from 30mg/L to 100mg/L (Yu et al., 2013; Zhou et al., 2022). It has been
reported that sulfate reduction process is one of the important ways of anaerobic
metabolism of organic matter in freshwater lakes, and $\sum S^{2-}$ produced by sulfate
reduction process can mediate the iron reduction process (Jorgensen et al., 2019; Zhang



et al., 2020). SRB mainly uses $SO_4^{2-}$ as the electron acceptor to complete anaerobic
respiration, and the sulfur compounds produced by anaerobic metabolism are bound
with iron and so on, which are fixed in the sediments and form AVS on the surface of
sediments (Holmer et al., 2001; Chen et al., 2016). Therefore, with the input of
exogenous sulfur, sulfate reduction process produced $\sum S^{2-}$ will further promote iron
reduction in freshwater lakes.
In freshwater lakes, iron cycle affects the process of phosphorus cycle, and sulfur
cycle plays an important role in regulating iron cycle. Therefore, the cycle of iron, sulfur
and phosphorus in freshwater lakes is inseparable (Wu et al., 2019; Zhao et al., 2019).
Studies have shown that even when $SO_4^{2-}$ content was as low as 20 mg/L, the anaerobic
metabolism of organic substrates was still dominated by sulfate reduction. Therefore,
sulfate reduction process plays an important role in the lacustrine biochemical cycle
(Hansel et al., 2015). In the absence of cyanobacteria, sulfate reduction doesn't occur
even if the $SO_4^{2-}$ concentration is higher (Zhao et al., 2021). This is because the
accumulation and decomposition of cyanobacteria not only change the environment of
water body, but also release a large amount of organic matter, which provides the
necessary conditions for the circulation of iron, sulfur and phosphorus (Yan et al., 2017;
Melemdez-Pastor et al., 2019). Therefore, under the co-effect of the increase of $SO_4^{2-}$
and the cyanobacteria decomposition, the sulfate reduction process and the effect of
iron reduction process on endogenous phosphorus release from sediments need to be
further studied.
In this study, a series of different initial concentrations of $SO_4^{2-}$ were set according





to the variation trend of $SO_4^{2-}$ concentrations over the years and the possible rising trend
of eutrophic Lake Taihu. The effects of increased $SO_4^{2-}$ concentration and cyanobacteria
bloom on sulfate reduction coupled with the microbial processes were investigated. In
addition, the dynamic changes of $Fe^{2+}$ and $Fe^{3+}$ concentrations during iron reduction
were studied in order to reveal the effect of sulfate reduction on iron reduction. The
effects of increasing sulfate concentration and cyanobacteria outbreak on sulfur cycle,
iron cycle and phosphorus cycle were also comprehensively analyzed for elucidating
the phosphorus release dynamics to tracking the hidden promoter of cyanobacteria
bloom occurrence in eutrophic lakes. The findings may be benefit for evaluating the
effect of sulfate reduction in freshwater lakes and its impact on the promotion of iron
reduction and the release of endogenous phosphorus.
**2.Materials and methods**
*2.1 Sample collection and preparation*
Lake Taihu (31°24' 40" N, 120°1' 3" E), one of the largest eutrophic shallow lakes
in China, with an average depth of 2.4 m and an area of 2340 $m^2$ (Mao et al., 2021).
Samples of sediments and cyanobacteria were collected in July 2020. Sediments from
the west shoreline of the lake (31°24'45''N, 120°0'42''E) were collected using a
peterson mud picker. Cyanobacteria bloom scums were collected and concentrated by
sieving water through a fine-mesh plankton (250 meshes). All the sediment and
cyanobacteria samples were stored in an incubator with ice packs and delivered to the
laboratory immediately. The sediment samples were blended thoroughly, homogenized,
and sieved (100 mesh) to the polyethylene bag. The cyanobacteria samples were flushed



and centrifuged at 1500 r/min for 5 min by a CT15RT versatile refrigerated centrifuge
(China) and freezed drying by Biosafer-10A. Different gradient sulfate concentrations
were prepared from the high purity water and $Na_2SO_4$.
*2.2 Set-up of incubation microcosms*

To simulate the dramatical $SO_4^{2-}$ increase and cyanobacteria blooms of eutrophic

Lake Taihu, a series of microcosms were constructed in this study. According to the
ratio of surface sediments and the average water depth and the cyanobacteria
accumulation density of 2500 $g/m^2$ during the breakout of cyanobacteria blooms of
Taihu Lake, 100 g of sediment, 200 ml of water and 0.11 g of cyanobacteria powder
were added into each bottle (Zhang et al., 2020). Meanwhile, according to the change
trend of $SO_4^{2-}$ concentrations in Taihu Lake over the years (Yu et al., 2013), the $SO_4^{2-}$
concentrations in six microcosm systems were configured as: 30, 60, 90, 120, 150 mg/L,
and a control without $SO_4^{2-}$, respectively. The microcosm adopts anaerobic bottle
($\Phi$75mm, length 180mm, volume 500ml) as the reaction device. There are three
replicates in each $SO_4^{2-}$ concentration experimental group. Since the sampling method
of the experiment is destructive sampling, 17 anaerobic bottles need to be set for each
parallel group according to the setting of experimental sampling times, so there are 6 $\times$
3 $\times$ 17 anaerobic bottles in total. All the anaerobic bottles were placed in biochemical
incubator at a temperature of 25 °C. Each group was sampled 17 times on 1, 2, 3, 4, 5,
6, 7, 9, 11, 14, 18, 23, 28, 33, 38, 43 and 48 d. The water, gas and soil samples were
collected by destructive sampling, three anaerobic bottles were collected in each group.
A part of sediment was used for microbe determination and kept in a refrigerator at -

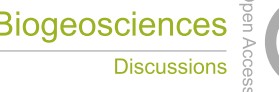

80 °C, and the rest sediment and other samples were kept at 0-4 °C for less than 24 h
before analysis.
*2.3 Chemical analytical methods*
All water column and pore-water samples were filtered through 0.45 μm Nylon
filters prior. Dissolved total phosphorus (DTP) was determined by colorimetry after
digestion with $K_2S_2O_8$+NaOH (Ebina et al., 1983). Water DO, oxidation and reduction
potential (ORP) were measured using calibrated probes (MP525, China) during
destructive sampling. The $SO_4^{2-}$ and $\sum S^{2-}$ were detected using the turbidimetric
(Tabatabai et al., 1974), methylene blue (Cline et al., 1969). Acid volatile sulfate (AVS),
the $\sum S^{2-}$ combined with metal ions formed compounds in sediments, was determined
by zinc cold diffusion method (Hsieh et al., 1997). $Fe^{2+}$ and $Fe^{3+}$ was determined by
colorimetrical (Phillips et al., 1987). The sediment total phosphorus (TP) was extracted
and determined by coloimetry (Ruban et al., 2001).
*2.4 Quantification of SRB in sediments*
In order to confirm the changes of sediment SRB in the microcosms, RT-QPCR
technologies were used to determine the cell copy numbers of MPA and SRB on 0,7
and 38 d in the sediments.
The sediment samples were collected and frozen at -80 °C in an ultra-low
temperature freezer. The E.Z.N.A. ®Soil DNA Kit (Omega Bio-Tek, Norcross, GA,
USA) was used to extract the total genomic DNA from each soil sample according to
the manufacturer's instructions. Nucleic acid quality and concentration were
determined by 1% agarose gel electrophoresis and NanoDrop 2000 UV





spectrophotometer (Thermo Scientific, USA), respectively.

SRB in sediments were quantified using the quantitative polymerase chain

reaction (qPCR) method. The qPCR with primer sets targeting DSR1F+ (5'-
ACSCACTGGAAGCACGGCGG-3') and DSR-R (5'-GTGGMRCCGTGCAKRTT
GG-3') were used for the SRB in this study. The q-PCR experiments were performed
on a ABI7300 q-PCR instrument (Applied Biosystems, USA) using ChamQ SYBR
Color qPCR Master Mix as the signal dye. Each 20 μL reaction mixture contained 2 μL
of the template DNA and 16.5 μL of ChamQ SYBR Color qPCR Master Mix. Standard
curves for each gene were obtained by the tenfold serial dilution of standard plasmids
containing the target functional gene. All operations were followed the MIQE
guidelines.
*2.5 Statistical analysis*

The Statistical Package of the Social Science 18.0 (SPSS 18.0) was used for

statistical analysis. The one-way analysis of variance (ANOVA) and correlation
analysis was carried out using bivariate correlations analysis.

**3.Results**
*3.1 $Fe^{2+}$ and $Fe^{3+}$ dynamics in overlying water*

The concentration variations of $Fe^{2+}$ and $Fe^{3+}$ in overlying water during the

incubation was presented in Fig.1. In the treatment without $SO_4^{2-}$, they increased
continuously to 9.68 mg/L and 10.15 mg/L, respectively. The concentration of $Fe^{3+}$ in
the remaining five treatments decreased at the beginning and then increased to keep



stable. The higher the initial sulfate concentration was, the lower the final $Fe^{3+}$
concentration displayed. In the initial 150 mg/L $SO_4^{2-}$ concentration treatment, the final
$Fe^{3+}$ concentration was the lowest of 7.7 mg/L. The $Fe^{2+}$ concentration in the five
treatments supplemented with $SO_4^{2-}$ decreased significantly from 11 d to 23 d, and then
increased to a stable level. The final concentration of $Fe^{2+}$ also showed a negative
correlation with the initial concentration of $SO_4^{2-}$. In the initial 30 mg/L $SO_4^{2-}$
concentration treatment, the final $Fe^{2+}$ concentration was the highest of 7.07 mg/L.

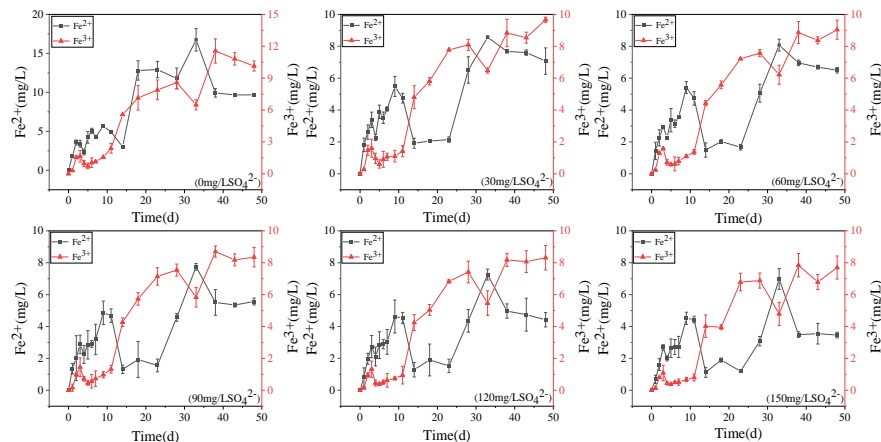


Figure 1. The concentration variations of $Fe^{2+}$ and $Fe^{3+}$ in the water column during the
incubation
*3.2 $SO_4^{2-}$ and $\sum S^{2-}$ dynamics in overlying water*

All treatments had obvious sulfate reduction reaction, and the concentration of

$SO_4^{2-}$ decreased greatly except for the treatment without adding $SO_4^{2-}$ (Fig.2). The
higher the initial sulfate concentration was, the faster the sulfate reduction rate in the
initial stage exhibited (Tab.1). In the treatment with initial $SO_4^{2-}$ concentration of 150
mg/L, the sulphate reduction rate was 39.68 mg/L·d, while it was only 9.39 mg/L·d in



the 30 mg/L $SO_4^{2-}$ treatment. The sulfate reduction rate at the beginning of other
treatments was also positively correlated with the initial $SO_4^{2-}$ concentration.
The higher the initial $SO_4^{2-}$ concentration was, the higher the maximum
concentration of $\sum S^{2-}$ was. In the treatment with initial $SO_4^{2-}$ concentration of 30 mg/L,
the lowest concentration was 2.93 mg/L on the 5th day. However, the lowest $SO_4^{2-}$
concentration appeared on the 23rd day was 1.18 mg/L in the treatment with initial
$SO_4^{2-}$ concentration of 150 mg/L. The maximum concentration of $\sum S^{2-}$ was positively
correlated with the initial $SO_4^{2-}$ concentration. In the initial $SO_4^{2-}$ concentrations of 30,
60, 90, 120 and 150 mg/L $SO_4^{2-}$ treatments, the highest $\sum S^{2-}$ concentrations at 7 d were
0.14, 0.61, 1.14, 1.55, 2.15, and 3.15 mg/L, respectively.
Table 1. Sulphate reduction rate in the water column with different initial $SO_4^{2-}$
concentrations

| Time(d) Groups | 0 | 7 | 38 |
|---|---|---|---|
| 0 mg/L$SO_4^{2-}$ | - | - | - |
| 30 mg/L$SO_4^{2-}$ | 9.39 | 0.74 | 0.05 |
| 60 mg/L$SO_4^{2-}$ | 9.44 | 2.84 | 0.07 |
| 90 mg/L$SO_4^{2-}$ | 28.02 | 4.98 | 0.11 |
| 120 mg/L$SO_4^{2-}$ | 30.89 | 19.45 | 0.11 |
| 150 mg/L$SO_4^{2-}$ | 39.68 | 10.42 | 0.21 |

* The units of sulphate reduction rate were mg/L·d





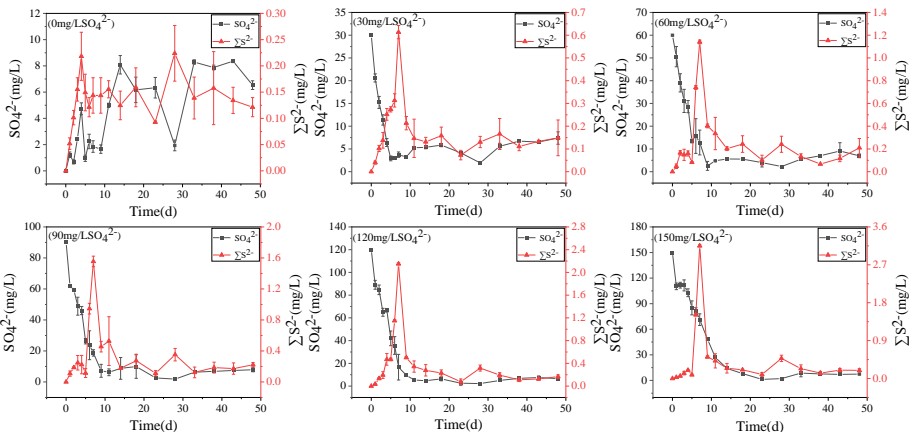

Figure 2. The concentration variations of $SO_4^{2-}$ and $\sum S^{2-}$ in the water column during the incubation

*3.3 TP dynamics in overlying water and sediments*

The dynamics of DTP concentrations in overlying water during the incubation was presented (Fig.3 left). The concentrations of DTP in overlying water were positively correlated with the initial $SO_4^{2-}$. The higher the initial concentrations of $SO_4^{2-}$ were, the higher the concentrations of DTP in overlying water were. On 11 day, DTP in overlying water continued to rise and then kept stable. The highest DTP concentration was 2.08 mg/L in the treatment with initial $SO_4^{2-}$ concentration of 150 mg/L, while the highest DTP concentration was 0.36 mg/L in the treatment without $SO_4^{2-}$ addition.

The concentrations of TP in the sediments increased significantly in all treatments with the cyanobacteria decomposition in the initial stage (Fig.3 right). Among of all treatments, on 9th day, the highest concentration of TP in the sediments was 887.69 mg/kg in the treatment with initial $SO_4^{2-}$ concentration of 0 mg/L. After 23 days, TP in the sediments decreased significantly and then stabilized. During cyanobacteria decomposition and sulfate reduction, the concentrations of TP in all treatments





negatively correlated with the initial $SO_4^{2-}$ concentration. The final TP concentration

was 448.92, 335.32, 321.56, 259.32, 238.56 and 227.21 mg/kg, respectively in all

treatments.

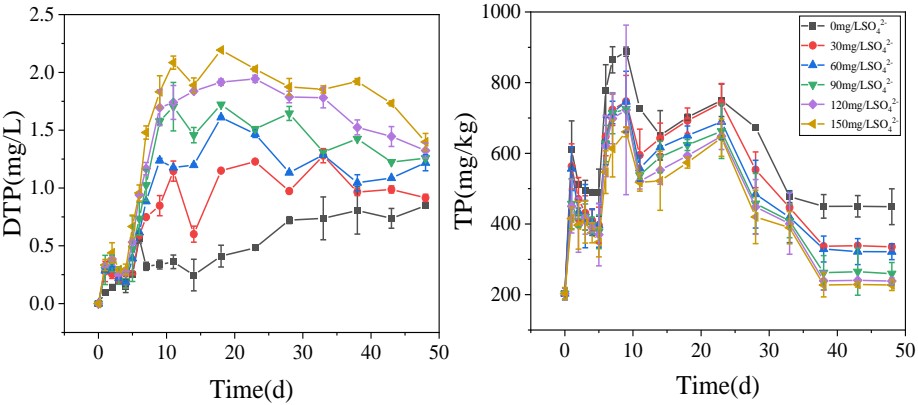

Figure 3. The concentrations of TP in the overlying water (left) and sediments (right)

during the incubation

*3.4 AVS dynamics in the sediments*

The concentrations of AVS in the sediments were positively correlated with the

initial $SO_4^{2-}$ concentrations. With the increase of TP in overlying water, the AVS in the

sediments also increased steadily and reached the peak on the 11st days. In the treatment

with initial $SO_4^{2-}$ concentration of 0, 30, 60, 90, 120 and 150 mg/L, the highest

concentration of AVS in the sediments were 7.21, 7.99, 8.54, 8.99, 9.34 and 11.11

mg/kg, respectively.






Figure 4. The concentration of AVS in the sediments during the incubation
*3.5 SRB dynamics in the sediments*

During the decomposition of cyanobacteria, SRB abundance significantly changed.

In the initial stage, the SRB abundance was $1.09*10^8$ copies/g and the final value was
positively correlated with the initial $SO_4^{2-}$. On 7 d, SRB of all treatments showed a
downward trend compared with the initial value, and there was no significant difference
in SRB values between each treatment. On 38 d, except for the initial $SO_4^{2-}$
concentrations of 0 and 30 mg/L, SRB increased significantly in other treatments.
Table 2. Copy numbers of the *dsrB* gene of SRB in the sediments during the incubation

| Time Groups | 0d | 7d | 38d |
|---|---|---|---|
| 0 mg/L$SO_4^{2-}$ | $1.09*10^8$ | $5.81*10^7$ | $6.65*10^7$ |
| 30 mg/L$SO_4^{2-}$ | $1.09*10^8$ | $6.13*10^7$ | $7.71*10^7$ |
| 60 mg/L$SO_4^{2-}$ | $1.09*10^8$ | $7.61*10^7$ | $1.15*10^8$ |
| 90 mg/L$SO_4^{2-}$ | $1.09*10^8$ | $7.87*10^7$ | $1.31*10^8$ |
| 120 mg/L$SO_4^{2-}$ | $1.09*10^8$ | $7.99*10^7$ | $1.49*10^8$ |
| 150 mg/L$SO_4^{2-}$ | $1.09*10^8$ | $8.23*10^7$ | $1.91*10^8$ |

* The units of SRB were copies/g






## 4.Discussion


It is generally acknowledged that climate warming and exogenous nutrient input
are the important contributors to the occurrence of cyanobacteria blooms (Huisman et
al., 2004; Yan et al., 2017). However, in this study, we found that the dramatically
increasing $SO_4^{2-}$ concentration in eutrophic lakes is also a non-negligible promoter for
the self-sustaining of cyanobacteria blooms. In eutrophic lakes, the decomposition of
cyanobacteria consumed DO in the water, and formed strong anaerobic reduction
conditions (Fig.S1). Cyanobacteria released large amounts of organic matter during
their decay and decomposition (Fig.S2), which promoted microbial growth (Tab. 2) and
ultimately promoted anaerobic reduction of sulfur and iron (Holmer et al., 2001). Fe-P
was desorbed to from free $Fe^{3+}$, which was reduced to $Fe^{2+}$ in anaerobic environments
(Fig.1). Free $Fe^{2+}$ combined with $\sum S^{2-}$ which generated by sulfate reduction and
eventually formed AVS fixed in the sediments (Fig.4), and phosphorus was released
from the sediments (Fig.3). Therefore, with increasing $SO_4^{2-}$ concentrations in
eutrophic lakes, the influence of sulfate reduction on phosphorus release is worth
further investigation.
Sulfur and iron in eutrophic lake sediments are directly related to iron and
phosphorus, and sulfur and phosphorus are also closely linked to bridges under the
action of iron (Zhang et al., 2020). Therefore, with the increase of $SO_4^{2-}$ concentration
in eutrophic lakes, the effect of sulfate reduction on phosphorus release from sediments
may be more important than previously recognized (Pester et al., 2012). Sulfate



reduction driven by SRB is an important organic metabolism pathway in natural
systems. During the sulfate reduction process, $SO_4^{2-}$ is an electron acceptor and its
concentration variation can significantly affect the sulfate reduction rate (Holmer et al.,
2001; Nakagawa et al., 2012). During sulfate reduction, $SO_4^{2-}$ is reduced to $\sum S^{2-}$ by
acquiring the electrons supplied by SRB oxidation, hence SRB plays an important role
in sulfate reduction (Sela-Adler et al., 2017). In the case of increased SRB abundance
(Tab. 2) and increased $SO_4^{2-}$ concentration, the sulfate reduction reaction was enhanced.
The $SO_4^{2-}$ concentration in the overlying water decreased significantly accompanied by
a temporary increase in $\sum S^{2-}$ (Fig.2). The highest concentrations of $\sum S^{2-}$ also increased
with the initial $SO_4^{2-}$ concentrations (Fig.5a). Interestingly, the $\sum S^{2-}$ decreased rapidly
after day 10 to almost zero at the end (Fig.2). This may result from the two keys: (a)
hydrogen sulfide overflows from the incubator; (b) sulfide migrates downward, and
combines with other substances in the sediment and is immobilized (Zhang et al., 2020).
In this study, TP in the overlying water has a significant positive correlation with the
initial $SO_4^{2-}$ concentrations ($R^2 = 0.96$; Fig3). The classical theory holds that iron
reduction by microorganisms leads to the release of iron-bound phosphorus in the
anaerobic layer of sediments, and when the formed $Fe^{2+}$ enters the aerobic water layer,
it is oxidized by $Fe^{3+}$ and bound to phosphorus again (Roden et al., 2006; Chen et al.,
2016). When the sulfate reduction process mediates the iron reduction process, the
released $Fe^{2+}$ combines with the product $\sum S^{2-}$ of sulfate reduction to form Fe-S, thus
weakening the reoxidation process of $Fe^{2+}$, and increasing the release of phosphorus
(Mort et al., 2010; Zhao et al., 2019). Therefore, with the increase of $SO_4^{2-}$





concentrations in eutrophic lakes, it significantly promoted the release of endogenous
phosphorus from the sediments.

Although from a thermodynamic point of view, iron reduction should take

precedence over sulfur reduction (Han et al., 2015). However, due to chemical kinetics,
sulfur reduction occurs before iron reduction, resulting in the simultaneous appearance
of $\sum S^{2-}$ and iron oxides (Han et al., 2015; Hansel et al., 2015). This is consistent with
the concentration variation of iron and sulfur in this study (Fig.1-3). It has been reported
that iron cycles in the water body will produce an intense response to the accumulation
of sulfide, that is, sulfate reduction can promote iron reduction (Friedrich et al., 2014;
Zhang et al., 2020). $\sum S^{2-}$ is the final product of sulfate reduction, which is toxic to
microorganisms and easy to combine with heavy metals such as $Fe^{2+}$ to form AVS in
lake sediments (Holmer et al., 2001). In this study, the concentration of AVS showed a
significant positive correlation with the initial concentration of $SO_4^{2-}$ (Fig. 4, 5b), which
was consistent with the highest concentration of $\sum S^{2-}$ observed in the overlying water
(Fig. 2, 5c). The concentrations of $Fe^{2+}$ and $Fe^{3+}$ in the overlying water increased
significantly, and $Fe^{2+}$ significantly decreased in the middle of the incubation (Fig. 1),
suggesting that $Fe^{2+}$ reduced by sulfate can be combined with the product $\sum S^{2-}$ (Fig. 2).
These results consistent with the trend that AVS in the sediments reached a peak after
11 days and $\sum S^{2-}$ in the water decreased rapidly after 9 days and remained at a lower
concentration (Fig. 2, 3). The reason for this phenomenon may be the formation of Fe-
S compounds that is finally fixed in the sediments (Zhao et al., 2019).

The $\sum S^{2-}$ mediated iron chemical reduction may lead to more environmental





effects, such as phosphorus mobilization (Zhang et al., 2020). In this study, the
concentration of $Fe^{2+}$ in the treatment without $SO_4^{2-}$ continued to rise, and was up to
the highest concentration among all treatments (Fig. 1). In contrast, the concentrations
of TP in the treatment without $SO_4^{2-}$ showed the lowest concentration among all
treatments (Fig. 1, 5a). This is caused by $Fe^{2+}$ and $Fe^{3+}$ recombining with phosphorus
and being immobilized in the sediments (Wu et al., 2019). In general, iron combines
with phosphorus to form siderite ($FePO_4 \cdot 2H_2O$) and blue iron ($Fe_3(PO_4)_2 \cdot 8H_2O$) and is
bound to the sediments (Taylor et al., 2011). However, when precipitation or reduction
separates iron from iron phosphate minerals, phosphorus bound to iron is released (Gu
et al., 2016).

In order to further elucidate whether the increasing $SO_4^{2-}$ concentrations in

overlying water result in the self-sustaining of eutrophication in shallow lakes, a
conceptual diagram was put forward (Fig. 6). It has been accepted that exogenous
nutrient inputs and climate warming have positive effects on the breakout of
cyanobacteria blooms. With the continuous input of exogenous sulfur, the $SO_4^{2-}$
concentration in the lake water increases significantly. When cyanobacteria blooms
start to decay, the overlying water shifts from the aerobic state to the strong anaerobic
state, providing carbon source to promote the growth of microorganisms such as SRB.
The increasing $SO_4^{2-}$ concentrations provide the electron for the sulfate reduction
process, resulting in the sulfate reduction and the release of a large amount of $\sum S^{2-}$. The
$Fe^{2+}$ released from the iron reduction process is captured by $\sum S^{2-}$, and finally the
combination of iron and P was reduced, promoting the release of endogenous



phosphorus. Therefore, it is necessary to pay attention to the effect of enhanced sulfate
reduction on endogenous phosphorus release in eutrophic lakes.

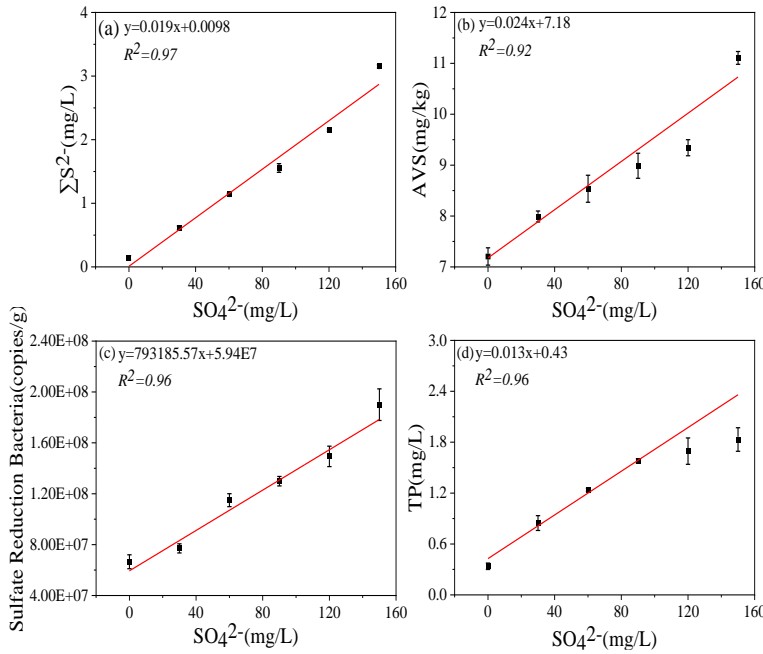


Figure 5. Correlation of initial $SO_4^{2-}$ concentrations with $\sum S^{2-}$ (a), AVS(b), Sulfate-
reducing bacteria (SRB) (c), TP (d) in the microcosm systems, respectively.

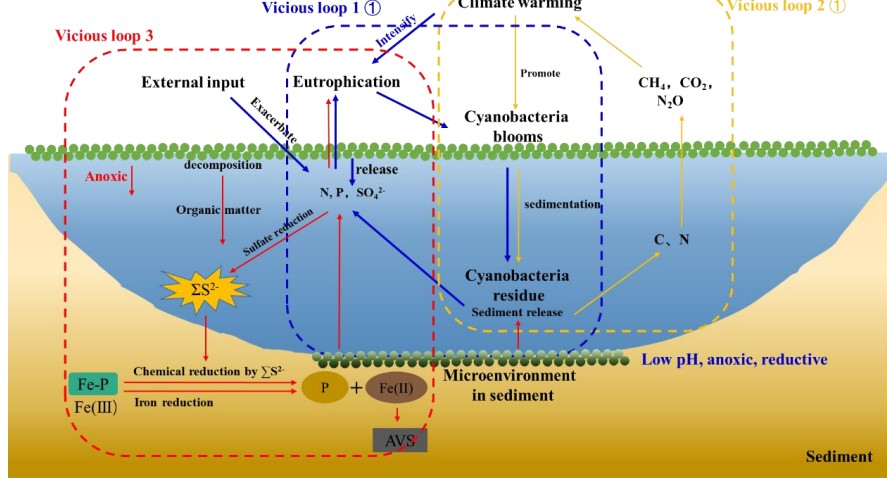




Figure 6. A simplified scheme of the relationship among climate warming, lake
eutrophication and cyanobacteria blooms in eutrophic lakes. Under climate warming
scenarios, extreme abiotic and biotic conditions facilitated the outbreak of
cyanobacteria blooms. After their collapse, the high amount of N, P, and C were
released into the overlying water and reacted with the eutrophication. Furthermore, a
large amount of $CH_4$ and $CO_2$ was produced and emitted to the atmosphere, contributing
to global warming of freshwater lakes (Yan et al. 2017). With the external sulfur input,
the concentration of $SO_4^{2-}$ increased significantly and sulfate reduction was enhanced.
The cyanobacteria decomposition created an anaerobic reduction environment, which
will promote iron reduction and sulfate reduction. The free $Fe^{3+}$ generated by Fe-P
desorption was reduced to $Fe^{2+}$ and combined with $\sum S^{2-}$ which produced by sulfate
reduction to form stable Fe-S in the sediments. Phosphorus was released from the
sediment into the overlying water. Therefore, there are three vicious loops between
cyanobacteria blooms occurrence, lake eutrophication and climate warming.

**5.Conclusion**
The dramatical increase of $SO_4^{2-}$ concentration was up to more than 100mg/L in
eutrophic lakes. There was a coupling relationship between sulfur, iron and phosphorus
cycles in lake ecosystems. Rapidly increasing sulfate concentration enhanced the
sulfate reduction to release of a large amount of $\sum S^{2-}$ mediated by the increasing
abundance of SRB with the adequate organic source from the decay processes of
cyanobacteria blooms. The iron reduction, in positive with initial sulfate concentration,



occurred with the cyanobacteria decomposition. The $Fe^{2+}$ released from the iron
reduction process was captured by $\sum S^{2-}$, and finally the combination of iron and P was
reduced, promoting the release of endogenous phosphorus. Therefore, except for
climate warming and excessive nutrients, the increasing sulfate concentration is proved
to be another hidden promoter of eutrophication in shallow lakes.

**Author contributions**
Xu Xiaoguang: designed and led the study. Zhou Chuanqiao, Peng Yu, Chen Li,
Yu Miaotong, Muchun Zhou, Xu Runze, Lanqing Zhang, Siyuan Zhang: performed the
investigation and analysed the samples. Zhou Chuanqiao and Peng Yu: wrote the
original draft with major edits and inputs from Xu Xiaoguang, Zhang Limin and Wang
Guoxiang.

**Competing interests**
The authors declare that they have no known competing financial interests or
personal relationships that could have appeared to influence the work reported in this
paper.

**Acknowledgements**
This work was supported by the National Natural Science Foundation of China
(No.42077294, 41877336, 41971043), the Cooperation and Guidance Project of
Prospering Inner Mongolia through Science and Technology (No.2021CG0037), the





National Key Research and Development Program of China (No.2021YFC3200304),
the Guangxi Key Research and Development Program of China (No.2018AB36010).

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
