# Peer review of "Rapidly increasing sulfate concentration: a hidden promoter of eutrophication in"

_Biogeosciences, 2022_

## Author Comment (AC1)

Reviewer 1:

In this manuscript, authors focused on the widespread increase of $SO_4^{2-}$ concentrations in eutrophic lakes, and they explored the driving mechanism on why increased sulfate concentration was a hidden promoter of eutrophication. Authors arranged a series of microcosms and measured chemicals including sulfate, $\sum S^{2-}$, AVS, $Fe^{2+}$, $Fe^{3+}$, TP and SRB in overlying water and sediments, which was comprehensive. This work provided new insight into the effects of sulfate reduction on the promotion of iron reduction and the release of endogenous phosphorus in freshwater lakes. Overall, I feel that the manuscript is easy to follow, and it is also interesting. However, several issues that need to be modified in this manuscript.

1.In this study, the sulfate concentration up to 150 mg/L was selected in the microcosms, however, such high sulfate might not occur in lakes. Authors need to add field data to prove this possibility.

Response:

Thanks for the reviewer's question. The sulfate concentration in the freshwater lakes increased significantly around the world [1]. Particularly for Lake Taihu, the sulfate concentration increased from 30 mg/L to 100 mg/L from 1960 to 2010[2]. In addition, it has been reported that the sulfate concentration will continue to increase in the future[3]. In this study, we set up the initial sulfate concentration according to the data from these studies and the future level in the eutrophic lakes.

[1] Holmer, M., Storkholm, P. Sulphate reduction and sulphur cycling in lake sediments: a review. Freshwater Biology, 2001, 46:431-451.

[2] Yu, T., Zhang Y., Wu, F.C., et al. Six-decade change in water chemistry of large freshwater lake Taihu, China. Environmental Science & Technology, 47(16): 9093-9101.

[3] Chen, M., Li, X.H., He, Y.H., et al. Increasing sulfate concentrations result in higher sulfide production an phosphorous mobilization in a shallow eutrophic freshwater lake. Water Research, 2016, 96: 94-104.

2.Line 85 "30mg/L to 100mg/L" lacks of space and the first mg/L need to be deleted. Please check throughout the manuscript.

Response:

We are sorry for our negligence. We have carefully checked and revised these errors throughout the manuscript.

3.Line 141 "0.11g if cyanobacteria powder were added into each bottle" What is the purpose of using cyanobacteria powder instead of fresh cyanobacteria? They have completely different ecological effects.

Response:

Thanks for your professional questions. In this study, we only considered the decomposition process of cyanobacteria, not the decay process. Therefore, we used the cyanobacteria powder instead of fresh cyanobacteria.

4.Line 141 "200 ml of water", water from Lake Taihu or prepared water in laboratory? How did you deal with it? Please explain it clear.

Response:

Special thanks to reviewer for your high perspicacity. In this study, the "200 ml of water" was prepared in laboratory, since the water from lake Taihu has the high concentration of sulfate[1]. It will affect the setup of microcosms for the initial sulfate concentration from 30 to 180 mg/L.

[1] Yu, T., Zhang Y., Wu, F.C., et al. Six-decade change in water chemistry of large freshwater lake Taihu, China. Environmental Science & Technology, 47(16): 9093-9101.

5.Lines 147-148 "Since the sampling method of the experiment is destructive sampling" what was "destructive sampling"? After sampling, how can you guarantee the stable anaerobic environment?

Response:

Thanks for the reviewer's question. In this study, we used the method of destructive sampling. At the beginning of the experiment, we set up a time series microcosms including 1, 2, 3, 4, 5, 6, 7, 9, 11, 14, 18, 23, 28, 33, 38, 43 and 48 d. At each time point of sampling, a few of anaerobic bottles were opened for testing, which ensured the anaerobic environment for other bottles.

6.Tab.1 and Tab.2: Why use the sampling data at 7 and 38 d? Why not the whole incubation? Please provide the reasons.

Response:

Thanks for the meaningful question. On 7 d, the TP in the overlying water was up to the highest concentration with the highest reduction rate of sulfate. Although the whole incubation lasted 48 days, all elements (S, Fe, P) in the anaerobic bottles remained stable after 38 d. Therefore, we used the sampling data at 7 and 38 d.

7.Line 311 "When the sulfate reduction process mediates the iron reduction process…" How can we confirm the occurrence of iron reduction or sulfate reduction? Authors need to explain the process and make it clear.

Response:

Thanks for your constructive suggestions. In this study, cyanobacteria decomposition formed the anaerobic environment in the overlying water which was an important factor for the occurrence of iron reduction and sulfate reduction. For iron reduction, the concentrations of $Fe^{3+}$ and $Fe^{2+}$ increased significantly in the overlying water. However, due to the iron reduction, the $Fe^{3+}$ concentration was lower than the $Fe^{2+}$ concentration. For sulfate reduction, the sulfate concentration in the overlying water decreased significantly and the concentrations of $\sum S^{2-}$ and AVS increased. Therefore, we can confirm the occurrence of iron reduction and sulfate reduction in this study.

8.A thorough byproducts investigation might be required to show the change and shift of oxidation-reduction processes.

Response:
Thanks for the reviewer's good suggestion. In this manuscript, we showed the dynamic changes of iron and $\sum S^{2-}$ concentrations in the overlying water, and the AVS concentration and SRB abundance in the sediment. These were important byproducts during oxidation-reduction processes.

9.Sulfate addition would affect the microbial diversity and cause the increase in SRB. SRB played an important role of sulfate reduction. However, there are no data to report these results.

Response:
As the reviewer has pointed it out, the sulfate addition affected the microbial process including SRB, actually, we have showed the dynamic changes of the SRB abundance on 0, 7 and 38 d in Table 2.

10.What is the minimum TOC for the occurrence of sulfate reduction and iron reduction for TP release? According to the discussion, lines 279-280, "Cyanobacteria released large amounts of organic matter during their decay and decomposition", the TOC might come from the cyanobacteria bloom. This indicates that some other carbon and nutrient sources are required to simulate the cyanobacteria bloom. Please clarify this description to prove your point "sulfate concentration increased was a hidden promoter of cyanobacteria bloom. "

Response:
Thank you for your important question. In this study, we focus on the endogenous phosphorus release from sediments. Cyanobacteria decomposition released phosphorus and the cyanobacteria biomass remained equal at the initial stage in all anaerobic bottles, however, the phosphorus concentration in the overlying water showed positive correlation with the initial sulfate concentration. The phosphorus concentration in the sediment showed the negative correlation with the initial sulfate concentration. These results clarified that the sulfate concentration promoted the endogenous phosphorus released from sediment to overlying water.

---

## Author Comment (AC2)

Reviewer 2:

The manuscript by Zhou et al. investigated the effects of different levels of sulfate concentrations and sulfate reduction on P mobility and release subjected to cyanobacteria decomposition. This work is interesting and the authors found a new contributing pathway of eutrophication in lakes. However, there are several problems still need to be revised especially in the experimental design and discussion sections.

1.The accumulation and decay of cyanobacteria in eutrophic lakes might change the physical and chemical environments of water body and form anaerobic reduction conditions. However, cyanobacteria decomposition also released a large amount of phosphorus. This has implications for determining how much phosphorus was released from the sediments, how does the authors solve this problem.

Response:

Thanks for the reviewer's good problem. In this study, the cyanobacteria powder biomass remained equal at the initial stage in all anaerobic bottles. During the incubation, the phosphorus concentration in the overlying water showed positive correlation with the initial sulfate concentration. The phosphorus concentration in the sediment showed the negative correlation with the initial sulfate concentration. These results indicated that sulfate concentration promoted the endogenous phosphorus released from sediment to overlying water.

2.Microorganisms play an important role in the biogeochemical cycle of lakes. The increased of sulfate concentrations will affect the abundance and activities of microorganisms. More data are needed to report these results in the discussion section.

Response:

Thanks for the reviewer's suggestion. The increase of sulfate concentration promoted the increase of abundance and activity of SRB. In this study, we have showed the dynamic changes of the SRB abundance in Table 1. We will add more data about the abundance and activities of microorganisms in the discussion section.

3.The authors set up a series of sulfate concentrations from 0 to 180mg/L. However, some concentrations of sulfate were too high, therefore, some background data of sulfate concentration can be added in eutrophic lakes and to explain the role of sulfate concentration gradients in microsystems.

Response:

Thanks for the reviewer's kind remind. In recent studies, the sulfate concentration increases significantly in the freshwater lakes around the world[1]. Particularly for Lake Taihu, the sulfate concentration increased from 30 mg/L to 100 mg/L from 1960 to 2010[2]. In addition, it has been reported that the sulfate concentration will continue to increase in the future [3]. We will add more background data of sulfate concentration to explain the role of sulfate concentration gradients in microsystems.

[1] Holmer, M., Storkholm, P. Sulphate reduction and sulphur cycling in lake sediments: a review. Freshwater Biology, 2001, 46:431-451.

[2] Yu, T., Zhang Y., Wu, F.C., et al. Six-decade change in water chemistry of large

freshwater lake Taihu, China. Environmental Science & Technology, 47(16): 9093-9101.

[3] Chen, M., Li, X.H., He, Y.H., et al. Increasing sulfate concentrations result in higher sulfide production an phosphorous mobilization in a shallow eutrophic freshwater lake. Water Research, 2016, 96: 94-104.

4.Why use cyanobacteria powder instead of fresh cyanobacteria? What is the meaning of cyanobacteria powder? The cyanobacteria powder and fresh cyanobacteria may have different ecological effects.

Response:

Thanks for your professional questions. In this study, we only considered the decomposition process of cyanobacteria, not the decay process. Therefore, we used the cyanobacteria powder instead of fresh cyanobacteria.

5.Please explain destructive sampling's definition and the reason for choosing this method. Authors need to explain how to keep anaerobic environment during incubation.

Response:

Thanks for the reviewer's questions. In this study, we used the method of destructive sampling. At the beginning of the experiment, we set up a time series microcosms including 1, 2, 3, 4, 5, 6, 7, 9, 11, 14, 18, 23, 28, 33, 38, 43 and 48 d. Only one group needs to be taken out in each sampling period, therefore, the anaerobic state of other anaerobic bottles will not be destroyed.

6.Line 40, Line 128 "cyanobacteria bloom" please keep the form of the full manuscript consistent. Line 116 "cyanobacteria outbreak", Line 367-368 "outbreak of cyanobacteria". Please keep the form of the full manuscript consistent.

Response:

We are sorry for these mistakes. We will adjust the format to ensure the unity throughout the manuscript.

7.Fig.1, the coordinate of iron concentration should be consistent in different groups for comparison and observation.

Response:

Thank you so much for your valuable suggestions. In this manuscript, we maintained consistent iron concentration coordinates for the treatments with initial sulfate concentration of 30, 60, 90, 120, 150 and 180 mg/L in Fig.1. In the treatment with 0 mg/L sulfate, the iron concentration in overlying water was higher than other treatments. We will unify the coordinate of iron concentration for comparison and observation.

---

## Author Comment (AC4)

Reviewer 3:

This manuscript introduced a story on the driving process and mechanism of the rapidly rising $SO_4^{2-}$ concentrations as a crucial contributor to the eutrophication in shallow lakes. Authors successively demonstrated the massive production of $\Sigma S^{2-}$ and the enhancement of iron reduction under the condition of rapid increase of $SO_4^{2-}$. The $Fe^{2+}$ released from the iron reduction process was captured by $\Sigma S^{2-}$, and the combination of iron and P was reduced, promoting the release of endogenous phosphorus. This experiment is relatively complete and novel, only some formatting issues need to be adjusted.

1.The introduction provided few quantitative data. For example, L76-79, "$SO_4^{2-}$ concentration in seawater reaching 28mM" was mentioned to support the $SO_4^{2-}$ concentration was the important influence factor of sulfate, what is threshold and why not make a comparison? And how is it related to $SO_4^{2-}$ concentration in eutrophic lakes.

Response:

Thanks for your very professional comments here. It has been reported that the intensity of sulfate reduction is high in freshwater lakes [1], which is affected by many factors, including the increased of sulfate concentration, and the rapid reoxidation of sulfur [2]. We will continue to explore the threshold of the sulfate reduction in next experiments. In eutrophic lakes, the degree of the eutrophication shows a positive correlation between the $SO_4^{2-}$ concentration and the degree of eutrophication [3].

[1] Sandfeld, T., Marzocchi, U., Petro, C., Schramm, A., Risgaard-Petersen, N. Electrogenic sulfide oxidation mediated by cable bacteria stimulates sulfate reduction in freshwater sediments. The ISME Journal, 2020, 14(5): 1233-1246.

[2] Holmer, M., Storkholm, P. Sulphate reduction and sulphur cycling in lake sediments: a review. Freshwater Biology, 2001, 46:431-451.

[3] Yu, T., Zhang Y., Wu, F.C., et al. Six-decade change in water chemistry of large freshwater lake Taihu, China. Environmental Science & Technology, 47(16): 9093-9101.

2.L85 "the $SO_4^{2-}$ concentration in Lake Taihu has increased from 30mg/L to 100mg/L"
L146 "$\Phi$75mm, length 180mm, volume 500ml"
Kindly add a space between number and unit except % in the whole manuscript.

Response:

We are very sorry for the incorrect writing. We will add a space between number and unit throughout the manuscript.

3.L129 "a fine-mesh plankton (250 meshes)"Inaccurate units, "es" should be deleted.

Response:

Thanks for your careful remind. We will delete the "es" in Line 129.

4.L224 "the highest $\Sigma S^{2-}$ concentrations at 7 d were 0.14, 0.61, 1.14, 1.55, 2.15, and 3.15 mg/L, respectively." There is one more punctuation mark before "and".

Response:

We are grateful to the reviewer for pointing out this error. We will modify the sentence according to the reviewer's suggestion.

5.Table 1, Table 2 and Figure 2 "0 mg/L $SO_4^{2-}$" Kindly add a space between unit and " $SO_4^{2-}$"

Response:

Thanks for your careful review. We will add a space between number and unit in Table 1, Table 2 and Figure 2.

6.Figure 6, Part of the text is obscured.

Response:

Thank you so much for your valuable suggestions. With your valuable suggestions, we will revise the Fig. 6.

---

## Author Comment (AC5)

Reviewer 4:

The manuscript submitted by Zhou and colleagues explored the effect of sulfate reduction on phosphorus release from sediment. The authors constructed a series of microecosystems with different initial concentration of $SO_4{}^{2-}$, and explained the mechanism of promoting the release of endogenous phosphorus according to the changes of sulfur, iron and phosphorus during the cyanobacteria decomposition. This study proposed that the release of endogenous phosphorus was an important reason for maintaining lake eutrophication, which provided a new insight for lake management. While the topic is interesting and relevant for the journal, there are also some questions about the whole story that the author needs to answer and modify.

1.The authors described carefully the collection of samples required for experiments and the set-up of incubation microcosms in the section of "2. Materials and Methods". However, some photos of sample sites and schematic diagrams of experimental groups will be more convincing and straightforward.
Response:
Thanks for the reviewer's good suggestion. We will add the photos of sample sites and schematic diagrams of experimental groups in this manuscript.

2.L157-167. The chemical analytical methods involved in the manuscript need further introduced. Authors need to add further detail to describe the index test method involved in manuscript.
Response:
Thanks for the reviewer's valuable suggestion. From Line 157 to Line 167, we showed the chemical analytical methods in this study, and we will add more detail to describe the index test method according to the reviewer's suggestion.

3.During sampling of incubation microcosms, how to control the anaerobic and air pressure changes in the gas extraction process?
Response:
Thanks for your professional questions. In this study, we used the method of destructive sampling. At the beginning of the experiment, we set up a time series microcosms including 1, 2, 3, 4, 5, 6, 7, 9, 11, 14, 18, 23, 28, 33, 38, 43 and 48 d. At each time point of sampling, only one group needs to be taken out in each sampling period, therefore, the anaerobic environment of other anaerobic bottles will not be destroyed.

4.This study has been conducted for 48 days. The source of reference for this time should be indicated. Is it any value to assume that the experiment lasts longer?
Response:
Special thanks to the reviewer for your high perspicacity. Before the formal experiment, we did a preliminary experiment. We combined with the results of the preliminary experiment and the contents reported in the publications [1], and determined that the experiment lasted for 48 days. We will add the source of reference for this time in the manuscript. The experiment lasting longer is meaningful but unnecessary for this study.

The cyanobacteria powder was decomposed completely at 48 days, and the environment in the anaerobic bottles were in a relatively stable state. In addition, we observed that the phosphorus concentration kept stable.

[1] Yan, X.C., Xu, X.G., Wang, M.Y., Wang, G.X., Wu, S.J., Li, Z.C., Sun, H., Shi, A., Yang, YH. Climate warming and cyaobacteria blooms: Looks at their relationshiops from a new perspective. Water Reseaech. 2017, 125, 449-457.

5.Figure 1: It seems complicated. I suggest highlighting the main line of the article and adding some easy-to-understand symbols.

Response:

Thanks for your professional suggestions. We showed the dynamic changes of the iron concentration ($Fe^{2+}$, $Fe^{3+}$) in Figure 1. We will highlight the main line of the article and add some easy-to-understand symbols according to the review's suggestions.

6.L262. "During the decomposition of cyanobacteria, SRB abundance significantly changed."

Please show the result by statistical results.

Response:

Special thanks to reviewer for your high perspicacity. We will show the result by statistical results in Line 262.

7.This study discussed that expect for climate warming and external input, the release of endogenous phosphorus is also an important reason of eutrophic lake. Why didn't the authors determine its proportion of contribution and discuss the contribution rate of endogenous nutrients in a more detailed way in the manuscript?

Response:

Thanks for the reviewer's professional suggestion. To determine the proportion of contribution for endogenous phosphorus is out of the purpose of this study. In future experiments, we will consider the isotope tracer method to determine the contribution of endogenous phosphorus.

8.L279-281. "Cyanobacteria released large amounts of organic matter during their decay and decomposition, which promoted microbial growth and ultimately promoted anaerobic reduction of sulfur and iron (Holmer et al., 2001)." The authors obtained this result based on the results and references. But a detailed explanation of the biochemical process followed this sentence. Since the anaerobic reduction of sulfur and iron is quite complex, I suggest that more attention should be paid to the logic of the discussion here. Putting this sentence after the biochemical explanation will make the discussion clearer.

Response:

Thanks for the reviewer's questions. Cyanobacteria decomposition released a large amount of organic matter and formed the anaerobic environment which promoted the sulfate reduction [1]. We will modify the logic of this paragraph and add more discussion about the biochemical explanation.

[1] Holmer, M., Storkholm, P. Sulphate reduction and sulphur cycling in lake sediments:

a review. Freshwater Biology, 2001, 46:431-451.

9.In this manuscript, the results and discussion of microorganisms are insufficient. I suggest that the author can supplement more data to make the study more comprehensive.
Response:
Thanks for your comments. The increase of sulfate concentration promoted the increase of abundance and activity of SRB. In this study, we have showed the dynamic changes of the SRB abundance in Table 1. We will add more discussion about abundance and activities of microorganisms in the discussion section according to other studies.

10.This study indicated that the sulfate reduction promoted the release of endogenous phosphorus in eutrophic lakes. The authors may be able to compare this study with the non-trophic lakes in the middle and lower reaches of the Yangtze River.
Response:
Thanks for the reviewer's kind remind. It has been reported that the sulfate concentration in eutrophic lakes has a stronger reduction potential than that in non-eutrophic lakes, since the availability of organic matter is one of the important factors limiting the occurrence of sulfate reduction. We will compare this study with the non-trophic lakes in the middle and lower reaches of the Yangtze River and add more discussion.

11.Some of the outdated references should be replace with more recent one.
Response:
Thanks for the reviewer's valuable suggestion. We will replace the outdated references.

---

## Author Response (AR2)

**Response to reviewers**

Dear Editor and Reviewers:

Thank you for your letter and for the reviewers' comments concerning our manuscript entitled "Rapidly increasing sulfate concentration: a hidden promoter of eutrophication in shallow lakes" (bg-2022-77). Those comments are all valuable and very helpful for revising and improving our paper, as well as the important guiding significance to our researches. Taking account of reviewers' comments, we have revised and improved the manuscript. We hope our revisions meet with approval. Revised portion is marked with blue in the manuscript. The main corrections in the paper and the responses to the reviewers' comments are as follows:

**Associate Editor:**

1. In the methodology, you have mentioned the collection of gravity cores. What is the dimension of the core? Did you subsample the gravity core for the measurement of pore water ionic concentrations? How the sampling was done and what method was followed to extract sediment pore water?

Response:

Thanks for your professional questions. In this study, the surface sediment (0-20cm) was collected by a gravity core sampler with the length of 1.5 m and the diameter of 0.2 m. However, we did not measure the pore water ionic concentrations. In this study, we focused on the phosphorus released from sediments with the increased of $SO_4^{2-}$ concentrations. Therefore, we measured the AVS and phosphorus concentrations in bulk sediments rather than the pore water. We have modified the expression in *materials and methods* section.

2. I could not find sediment pore water data in the manuscript result section. Do you have the concentration data of pore water? Better to mention how AVS was extracted and quantified at least briefly. Just referencing may not be enough.

Response:

Thanks for the meaningful question. In this study, we did not measure the pore water as mentioned above. For the extraction and quantification of AVS in the sediment, we have added the schematic diagram and the methods from line 171 to 177 in the revised manuscript.